# Evaluation of Groundwater Quality for Irrigation in Deep Aquifers Using Multiple Graphical and Indexing Approaches Supported with Machine Learning Models and GIS Techniques, Souf Valley, Algeria

Mohamed Hamdy Eid [1,2,*], Mohssen Elbagory [3,4], Ahmed A. Tamma [5], Mohamed Gad [6], Salah Elsayed [7,*], Hend Hussein [8], Farahat S. Moghanm [9], Alaa El-Dein Omara [4], Attila Kovács [1] and Szűcs Péter [1]

1   Institute of Environmental Management, Faculty of Earth Science, University of Miskolc, 3515 Miskolc, Hungary
2   Geology Department, Faculty of Science, Beni-Suef University, Beni-Suef 65211, Egypt
3   Department of Biology, Faculty of Science and Arts, King Khalid University, Mohail 61321, Saudi Arabia
4   Agricultural Research Center, Department of Microbiology, Soils, Water and Environment Research Institute, Giza 12112, Egypt
5   Institute of Environmental Engineering, Faculty of Environmentsl Engineering and Geodesy, Wrocław University, 50-363 Wrocław, Poland
6   Hydrogeology, Evaluation of Natural Resources Department, Environmental Studies and Research Institute, University of Sadat City, Sadat City 32897, Egypt
7   Agricultural Engineering, Evaluation of Natural Resources Department, Environmental Studies and Research Institute, University of Sadat City, Sadat City 32897, Egypt
8   Geology Department, Faculty of Science, Damanhour University, Damanhour 22511, Egypt
9   Soil and Water Department, Faculty of Agriculture, Kafrelsheikh University, Kafr El-Sheikh 33516, Egypt
*   Correspondence: mohamedhamdy@science.bsu.edu.eg (M.H.E.); salah.emam@esri.usc.edu.eg (S.E.)

**Abstract:** Irrigation has made a significant contribution to supporting the population's expanding food demands, as well as promoting economic growth in irrigated regions. The current investigation was carried out in order to estimate the quality of the groundwater for agricultural viability in the Algerian Desert using various water quality indices and geographic information systems (GIS). In addition, support vector machine regression (SVMR) was applied to forecast eight irrigation water quality indices (IWQIs), such as the irrigation water quality index (IWQI), sodium adsorption ratio (SAR), sodium percentage (Na%), soluble sodium percentage (SSP), potential salinity (PS), Kelly index (KI), permeability index (PI), potential salinity (PS), permeability index (PI), and residual sodium carbonate (RSC). Several physicochemical variables, such as temperature (T°), hydrogen ion concentration (pH), total dissolved solids (TDS), electrical conductivity (EC), $K^+$, $Na^{2+}$, $Mg^{2+}$, $Ca^{2+}$, $Cl^-$, $SO_4{}^{2-}$, $HCO_3{}^-$, $CO_3{}^{2-}$, and $NO_3{}^-$, were measured from 45 deep groundwater wells. The hydrochemical facies of the groundwater resources were Ca–Mg–Cl/SO$_4$ and Na–Cl$^-$, which revealed evaporation, reverse ion exchange, and rock–water interaction processes. The IWQI, Na%, SAR, SSP, KI, PS, PI, and RSC showed mean values of 50.78, 43.07, 4.85, 41.78, 0.74, 29.60, 45.65, and −20.44, respectively. For instance, the IWQI for the obtained results indicated that the groundwater samples were categorized into high restriction to moderate restriction for irrigation purposes, which can only be used for plants that are highly salt tolerant. The SVMR model produced robust estimates for eight IWQIs in calibration (Cal.), with $R^2$ values varying between 0.90 and 0.97. Furthermore, in validation (Val.), $R^2$ values between 0.88 and 0.95 were achieved using the SVMR model, which produced reliable estimates for eight IWQIs. These findings support the feasibility of using IWQIs and SVMR models for the evaluation and management of the groundwater of complex terminal aquifers for irrigation. Finally, the combination of IWQIs, SVMR, and GIS was effective and an applicable technique for interpreting and forecasting the irrigation water quality used in both arid and semi-arid regions.

**Keywords:** hydrogeochemistry; reverse ion exchange; water quality indices; irrigation

## 1. Introduction

Groundwater resources are essential natural resources for a country's socioeconomic growth. However, agriculture consumes a very high percentage of the global groundwater [1]. These natural resources are dealing with a variety of problems that endanger their long-term viability, including the results of environmental issues, human impacts, and natural forces [2–6]. In general, these issues degrade the physical and chemical composition of groundwater, rendering it unfit for agriculture. Irrigation water availability is among the decisive variables for the increase of agricultural output in arid and semi-arid countries, both in terms of raising crop yields and expanding irrigated lands.

Semi-arid countries including Algeria are affected by water shortages associated with the degradation of the chemical composition and water quality due to the fact of the overexploitation of water resources. For instance, Algeria's Saharan groundwater is the second most important source for irrigation, industry, and drinking [7]. The population increase, agricultural production, and heavy industries in this area have resulted in a significant reduction in the aquifer system's water level. The aquifer in the northeast Algerian Sahara is one of the largest aquifers in the world, which consists of a Terminal Complex aquifer (CT) and a continental intercalary aquifer (CI) [8].

The groundwater production in the study area (Souf Valley) increased from 600 to 2120 $Mm^3$/y from 1970 to 2020. The groundwater extracted from the CT aquifer is utilized for both drinking and irrigation, with continuous increases in the number of operating wells (203 water wells in 2019) [9–11]. The regular observation and hydrochemical assessment of the groundwater's quality are necessary for the sustainable development and strategic planning of groundwater resources under this semi-confined aquifer due to the fact of the gradual decline in the groundwater resources' availability and quality [12–20].

Hydrogeochemistry studies of groundwater have commonly been utilized to evaluate and categorize the water quality. Therefore, some irrigation water's hydrochemical elements can have a detrimental influence on crop productivity and soil degradation [21]. The various parameters affecting water quality have already been evaluated by comparing to established values among many hydrochemical investigations to properly assess the quality of groundwater. This type of assessment cannot provide a complete description of the water quality for decision makers who need quick information, particularly when various water quality degraders exist at the same time. Application programming interfaces, such as the USSL diagram, Doneen, and Wilcox diagram, assist in determining the appropriateness of groundwater for irrigation use [22–24].

The IWQIs are derived from the chemical parameters that are effective methods to detect the water's suitability for irrigation by combining various water quality metrics into a single value that assists decision makers and managers [25–29]. Irrigation water quality (IWQ) for agricultural applications is routinely analyzed using a variety of indices and parameters based on Food and Agriculture Organization (FAO) criteria [30]. The eight indices, including the IWQI, Na%, SAR, SSP, PS, KI, PI, and RSC, have been utilized to categorize the appropriateness of water resources for irrigation in which the solutes concentration in the soil may lead to change in the soil permeability and crop productivity [24,31–35]. Several investigations have been undertaken to assess water quality for agricultural utilization through the integration of IWQIs and geographic information systems (GIS) technology, which enables the separation of quality zones for irrigation via creating thematic maps of groundwater quality [36–38].

Traditional irrigation water quality assessment methods are frequently costly and time consuming for agricultural producers, especially in developing countries. Machine learning (ML) implementations, such as SVMR, can address this issue by predicting and evaluating aquifer irrigation water quality indices based on chemical and physical parameters [39,40].

Moreover, new and cost-effective technologies for analyzing and predicting groundwater quality are required for long-term groundwater management strategies. As a consequence, prediction-based techniques in groundwater administration and management could be useful in resolving this issue. The prediction of water quality indices is critical for safe environmental management. Several deterministic models have previously been used in this domain in recent times [40–42].

However, because real-world natural ecosystems are commonly very difficult and complicated for these cutting-edge models, their statistical effectiveness is frequently poor. As a result, several techniques and methodologies for evaluating groundwater quality have been applied with positive outcomes. For quantifying and tracking groundwater quality, these methodologies involve index-based methods, statistical measures, and GIS techniques. Furthermore, it is important to determine the capabilities of ML models such as SVMR to forecast the different IWQIs in deep aquifers in the Algerian Desert using physical and chemical factors as input variables.

Therefore, this research study was conducted to (i) investigate the chemistry, types of groundwater, and the geochemical governing processes using physicochemical variables and imitative approaches; (ii) evaluate the groundwater appropriateness for irrigation purpose utilizing multiple irrigation water quality indices; and (iii) detect the precision of applying ML, especially SVMR models, to estimate the groundwater IWQI, SAR, Na%, PS, SSP, PI, KI, and RSC.

## 2. Materials and Methods

### 2.1. Site Descriptions and Hydrogeological Settings

The Souf valley research area is located in the Debila and El-Oued Districts of the Algerian Sahara's northeast. The Souf valley is located between longitude 6°40′00″/7°5′00″ E and latitude 33°12′00″/33°35′00″ N and has a population of approximately 900,000 [10]. The research location has a hot, dry summer environment, with evapotranspiration of 1224 mm/y. The sampling sites were largely concentrated in El-Oued areas and Debila areas, with some samples taken outside of these areas to examine the significant variations in the groundwater flow path (Figure 1).

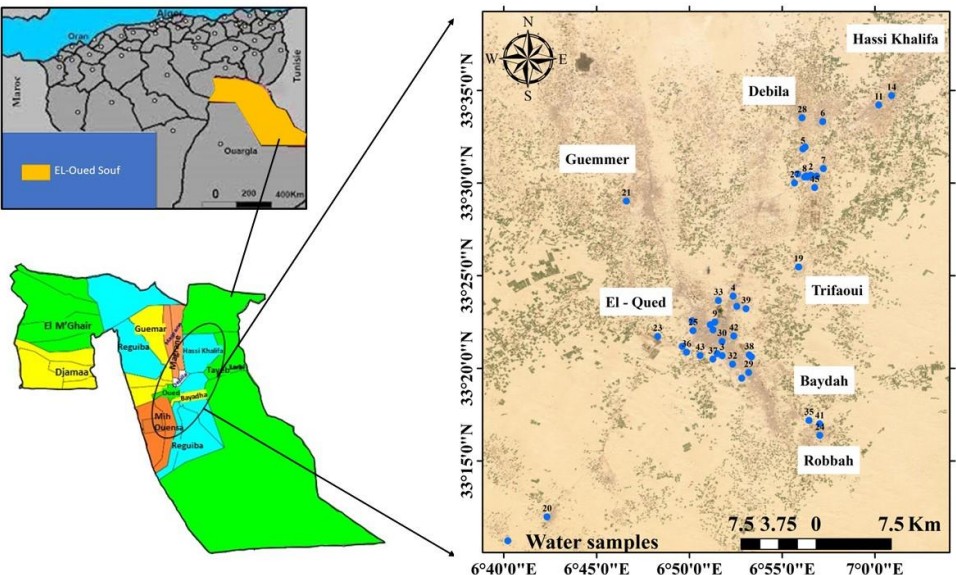

**Figure 1.** Groundwater samples and location map of Souf Valley.

The hydrogeological system of the northeastern Sahara of Algeria consists of three major aquifers, arranged from top to bottom as follow: shallow Quaternary aquifer, CT aquifer, and CI aquifer [43–45]. The CT aquifer consists of three formations (Figure 2): limestone and dolomite (Senonian–Eocene), gravel (lower Pontian), and sandstone (Mio–Pliocene).

The average thickness of the CT aquifer in the Souf Valley is approximately 300 m, with an average depth of 220 m. The groundwater's regional flow is from the southwest to the northeast. The Mih-Ouensa recorded the maximum piezometric level, which decreased in the direction of the El-Oued and Trifaoui [9]. The water level that was measured in 15 observation wells illustrates that most locations had significant drawdown in the water level due to the fact of over pumping of groundwater for irrigation. The regions of Trifaoui and El-Oued (Figure 3) were affected by the depletion in the piezometric head.

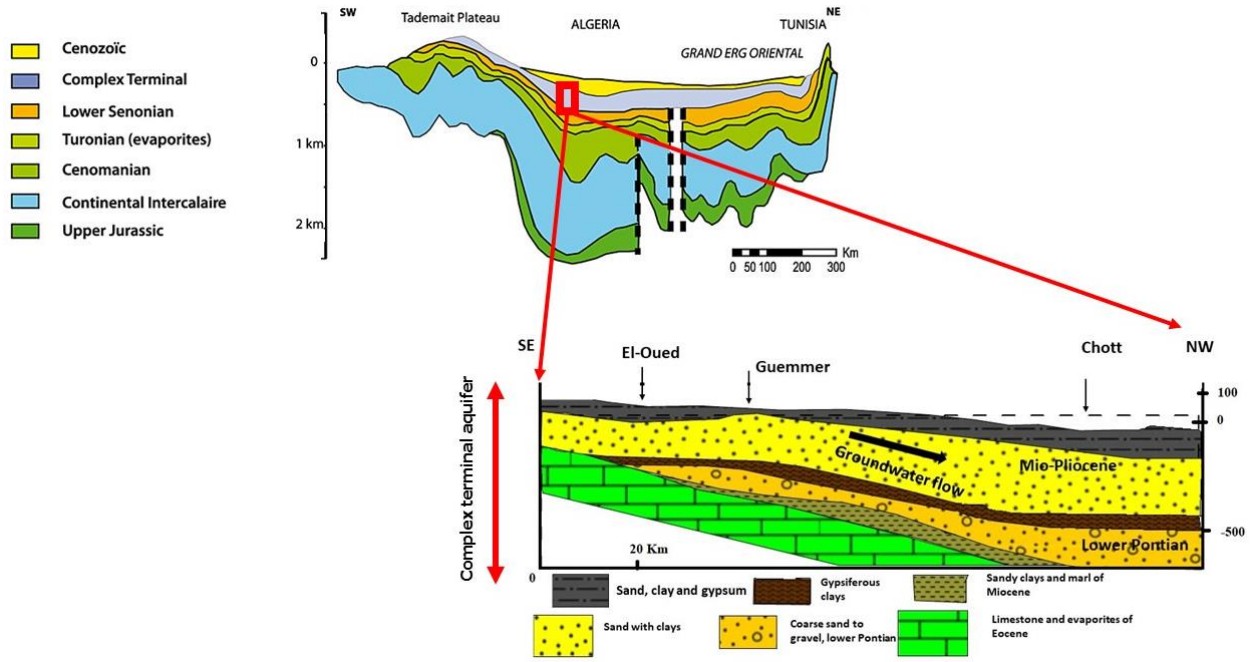

**Figure 2.** Hydrogeological cross-section of the CT aquifer [44,46,47].

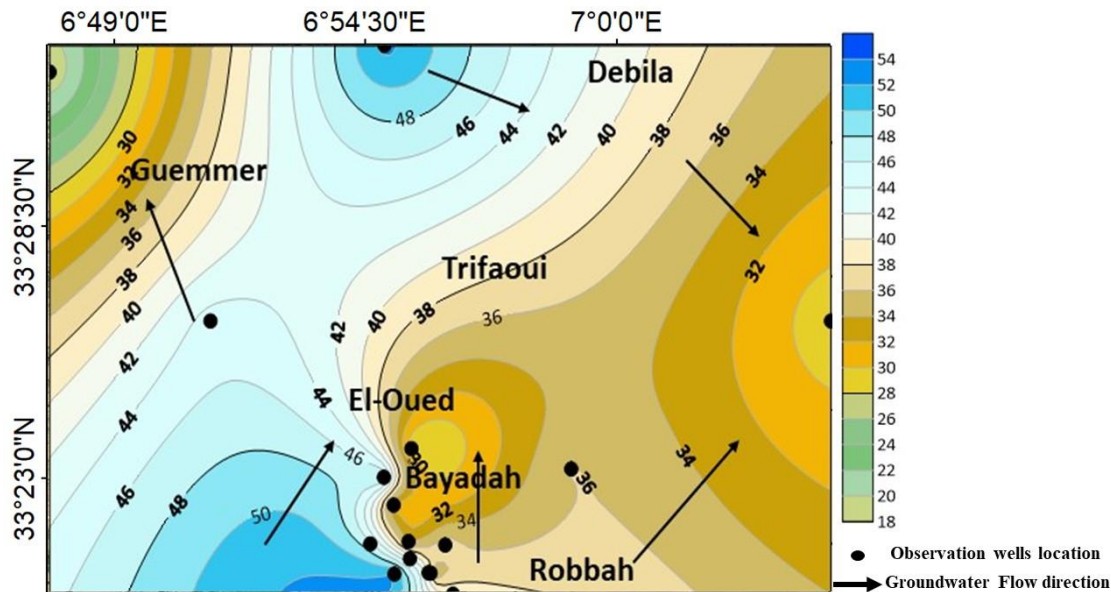

**Figure 3.** Groundwater flow direction from a Piezometric map of the CT aquifer.

### 2.2. Sampling and Analysis

Forty-five groundwater samples from wells with depths that varied from 139 to 526 m were collected during the year 2020 from the semi-confined Mio–Pliocene and Lower Pontian aquifers. The water samples were collected in acidified polyethylene bottles after

filtration for laboratory analysis. The T°, pH, TDS, EC, and ground elevation were all measured in the field. The pH and EC were measured using a portable multi-meter (HI 9829 type, YSI Professional Plus, Portugal). The major ions, $Ca^{2+}$, $Na^+$, $Cl^-$, $Mg^{2+}$, $K^+$, $HCO_3^-$, $SO_4^{2-}$, $CO_3^{2-}$, and $NO_3^-$, were analyzed in the laboratory. $NO_3^-$, $SO_4^{2-}$, and $Cl^-$ were measured using a HACH (DR2000 type, Loveland, CO, USA) spectrophotometer, while $Ca^{2+}$, $K^+$, and $Na^+$ were measured using a flame spectrophotometer (ELEX 6361, Eppendorf AG, Hamburg, Germany). $Mg^{2+}$ was analyzed using the complexometric method, while $CO_3^{2-}$ and $HCO_3^-$ were measured using the titrimetric method. The location map and spatial distribution map for the various parameters were created using quantum geographic information system (QGIS) and Surfer software's (Version 3.28.2). Graphical representations that included the Piper plot, Durov diagram, Gibbs plot, USSL diagram, Wilcox diagram, and binary plots were created using DIAGRAMMES software (version 6.77) and Excel (version 2010). According to Equation (1), the confirmation of the analytical error of the analyzed ions' concentration, in $meq/L^{-1}$, was cross-checked utilizing the charge balance error (CBE) within the limit of 5% [48]. The analytical processes were validated in regard to quality control by conducting adequate device calibration and evaluating the accuracy of the samples analyzed.

$$\text{CBE} = \left[ \frac{\sum \text{eq cations} - \sum \text{eq anions}}{\sum \text{eq cations} + \sum \text{eq anions}} \right] \times 100 \tag{1}$$

*2.3. Indexing Approach*

2.3.1. Irrigation Water Quality Indices (IWQIs)

The physicochemical data from the groundwater samples were used to generate the eight WQIs (Table 1).

**Table 1.** The formula and references of the IWQIs.

| Index | Formula | Reference |
|-------|---------|-----------|
| IWQI | $\sum_{i=1}^{n} Q_i W_i$ | [45] |
| Na% | $[(Na^{2+} + K^+)/(Ca^{2+} + Mg^{2+} + Na^{2+} + K^+)] \times 100$ | [46] |
| SAR | $\left( \frac{Na^+}{\sqrt{(Ca^{2+} + Mg^{2+})/2}} \right) \times 100$ | [47] |
| SSP | $[Na^{2+}/(Ca^{2+} + Mg^{2+} + Na^{2+})] \times 100$ | [46] |
| KI | $KI = Na^+/(Ca^{2+} + Mg^{2+})$ | [48] |
| PS | $Cl^- + (SO_4^{2-}/2)$ | [49] |
| PI | $[(Na^{2+} + \sqrt{HCO_3^-})/(Ca^{2+} + Mg^{2+} + Na^{2+})] \times 100$ | [49] |
| RSC | $(HCO_3^{2-} + CO_3^-)-(Ca^{2+} + Mg^{2+})$ | [50] |

Note: The IWQI was calculated in mg/L and the rest of the indices in meq/L.

2.3.2. Irrigation Water Quality Index (IWQI)

Using a non-dimensional scale, the IWQI measures from 0 to 100, which was computed in relation to variables such as EC, SAR, $Na^+$, $Cl^-$, and $HCO_3^{2-}$ [31,49], as per the equation:

$$\text{IWQI} = \sum_{i=1}^{n} Q_i W_i \tag{2}$$

where $W_i$ is the calculated weight of every variable, and $Q_i$ is the quality measurement value based on the permissible limits (Table 2).

$$Q_i = Q_{max} - \left( \frac{\left[ (X_{ij} - X_{inf}) \times Q_{imap} \right]}{X_{amp}} \right) \tag{3}$$

where $X_{ij}$ is each parameter's observed value, $X_{inf}$ is the value that correspond to the lower limit of the class, $Q_{imap}$ is the class amplitude, and $X_{amp}$ is the class amplitude to which the parameter belongs.

**Table 2.** Limiting values for the variables used in the computation of quality measurement (Qi).

| $Q_i$ | EC (μs/cm) | SAR | Na$^+$ (emp) | Cl$^-$ (emp) | HCO$_3{}^{2-}$ (epm) |
|---|---|---|---|---|---|
| 85–100 | $200 \leq EC < 750$ | $2 \leq EC < 3$ | $2 \leq Na < 3$ | $1 \leq Cl < 4$ | $1 \leq HCO_3 < 1.5$ |
| 60–85 | $750 \leq EC < 1500$ | $3 \leq EC < 6$ | $3 \leq Na < 6$ | $4 \leq Cl < 7$ | $1.5 \leq HCO_3 < 4.5$ |
| 35–60 | $1500 \leq EC < 3000$ | $6 \leq EC < 12$ | $6 \leq Na < 9$ | $7 \leq Cl < 10$ | $4.5 \leq HCO_3 < 8.5$ |
| 0–35 | $EC < 200$ or $EC \geq 3000$ | $SAR > 2$ or $SAR \geq 12$ | $Na < 2$ or $SAR \geq 9$ | $Cl < 1$ or $Cl \geq 10$ | $HCO_3 < 1$ or $HCO_3 \geq 8.5$ |

Finally, the value of Wi was computed via Equation (4):

$$W_i = \frac{\sum_{j=1}^{k} F_j A_{ij}}{\sum_{j=1}^{k} \sum_{i=1}^{n} F_j A_{ij}} \tag{4}$$

where F is the component 1's auto value; A is the degree to which factor j can explain parameter i, which is the number of physicochemical variables chosen by the model, ranging from 1 to n; and j is the number of factors to choose in IWQI, ranging from 1 to k.

2.3.3. Support Vector Machine Regression (SVMR)

In the current study, the SVMR models were applied to forecast the eight IWQIs. The eight IWQIs models were built by utilizing unscramble X software (version 10.2). The SVMR models used all of the selected parameters (Table 1) as the input data information to forecast the eight IWQIs as output data (Figure 4). The SVMR aims to compute a function from the supplied dataset (x, y) [20] in which x denotes the input vector (where x represents the water quality parameter), and y is the outcome (y represents the predicted IWQIs).

The following is a description of the SVMR function:

$$f(x) = \omega^T \varphi(x) + b \tag{5}$$

in which $f(x)$ is a representation of the output of the model, and $\varphi(x)$ is a representation of a nonlinear mapping function. The weight vector ($\omega$) and the bias ($b$) term should be optimized according the regularized function as follows:

$$\begin{cases} \min R(\omega, \xi, \xi^*, \varepsilon) = \frac{1}{2}\| \omega \|^2 + C\left[v\varepsilon + \frac{1}{l}\sum_{i=1}^{l}(\xi_i + \xi_i^*)\right] \\ subject to : y_i - \omega^T \varphi(x_i) - b \leq \varepsilon + \xi_i \\ \omega^T \varphi(x_i) + b - y_i \leq \varepsilon + \xi_i \\ \xi_i, \varepsilon \geq 0 \end{cases} \tag{6}$$

where $C$ is the correction parameter needed to counterbalance the overfitting and the model normalization component $\| \omega \|^2$; $\xi_i$ and $\xi_i^*$ are the positive slack variables. Utilizing Lagrange multipliers, the aforementioned SVR model is resolved.

$$\begin{cases} max\ R\left(a_i, a_i^*\right) = \sum_{i=1}^{l}\left(a_i^* - a_i\right) - \frac{1}{2}\sum_{i=1}^{l}\sum_{j=1}^{l}\left(a_i - a_i^*\right)\left(a_j - a_j^*\right)K\left(x_i, x_j\right) \\ subjective\ to : \sum_{i=1}^{l}\left(a_i - a_i^*\right) = 0 \\ 0 \leq a_i, a_i^* \leq \frac{C}{l} \\ \sum_{i=1}^{l}\left(a_i + a_i^*\right) \leq C.v \end{cases} \tag{7}$$

In this case, the kernel function is $K\left(x_i, x_j\right)$, and the positives Lagrange multipliers are $a_i$ and $a_i^*$, accordingly.

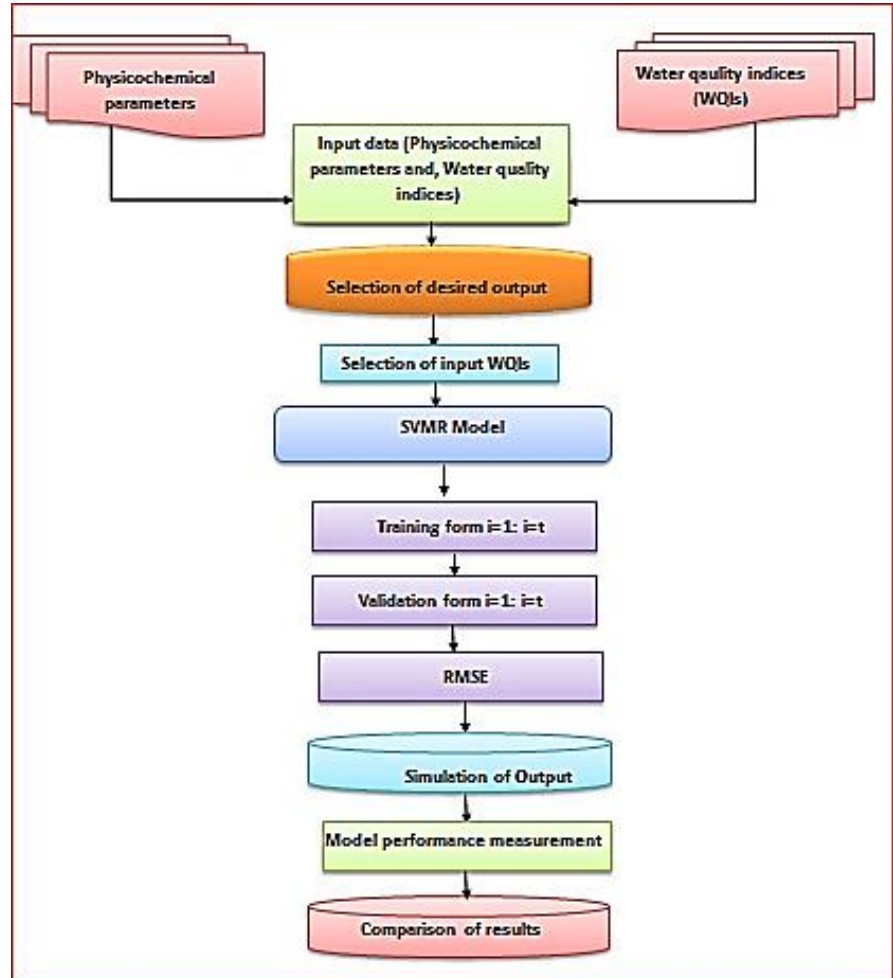

**Figure 4.** Diagram illustrating the SVMR methodology used in this study to forecast IWQI, Na%, SAR, SSP, KI, PI, PS, and RSC on the basis of the chosen chemical parameters.

After achieving the desired solution for the objective function, the SVM parameter models were ultimately established; thus, in the following regression, the regression formula was used to represent an input vector x.

$$f(x) = \sum_{i=1}^{l}(a_i^* - a_i)\, K(x_i\,,\, x_j) + b \tag{8}$$

Three metrics were utilized to evaluate how accurately the SVMR models predicted the six IWQIs: $R^2$ coefficient, RMSE, and equation slope.

Data Analysis and Processing

For the physicochemical characteristics and IWQIs, statistical analyses were performed using SPSS software, version 22 (SPSS Inc., Chicago, IL, USA). A Piper trilinear diagram [50] was created using Windows software (GWW, Lake Forest, IL, USA), version 1.30, to determine the hydrochemical evolution and water types according to cation and anion compositions. For anions and cations, a Gibbs diagram is usually used to depict the association with both the water chemistry and the aquifer metrics features using an Excel sheet [51,52]. For the graphical representations, different plots, including binary plots, Durov, Wilcox, and USSL diagrams, Excel and DIAGRAMMES software were used. Inverse distance weighting (IDW, San Diego, CA, USA) was utilized in ArcGIS 10.2 to determine the spatial distribution of the IWQIs.

## 3. Results and Discussion

### 3.1. Physicochemical Parameters of Groundwater

The physicochemical characteristics of the groundwater resources in the CT aquifer contributed significantly to the assessment of their quality and fitness for irrigation use, and they served as a useful method for determining the specific environmental issues, identifying patterns, and disseminating information concerning the groundwater resources, water quality, and geochemical processes. The following variables were used to categorize the groundwater fitness for agricultural purposes in the chosen aquifer. The chemical parameters, such as $T°$, EC, pH, TDS, $Na^+$, $K^+$, $Mg^{2+}$, $Ca^{2+}$, $Cl^-$, $HCO_3^-$, $SO_4^{2-}$, $CO_3^{2-}$, and $NO_3^-$, altered the soil productivity and soil quality. Table 3 shows the statistical methods of the main variables in the groundwater samples.

**Table 3.** Statistical analysis of the physical and chemical variables of the groundwater in the CT aquifer.

| Parameter | T °C | pH | EC | TDS | $K^+$ | $Na^+$ | $Mg^{2+}$ | $Ca^{2+}$ | $Cl^-$ | $SO_4^{2-}$ | $HCO_3^-$ | $CO_3^{2-}$ | $NO_3^-$ |
|---|---|---|---|---|---|---|---|---|---|---|---|---|---|
| | | | | | Deep Groundwater Aquifers, Algeria (*n* = 45) | | | | | | | | |
| Min. | 13.9 | 7.0 | 2640 | 1702 | 12.00 | 210.00 | 24.30 | 168.33 | 560.15 | 532.20 | 105.80 | 0.00 | 0.83 |
| Max. | 38.8 | 7.8 | 4360 | 2790 | 42.00 | 540.00 | 157.90 | 340.68 | 1127.40 | 840.27 | 195.20 | 1.03 | 31.53 |
| Mean | 26.6 | 7.4 | 3646 | 2342 | 33.30 | 364.38 | 109.70 | 258.98 | 822.25 | 697.53 | 135.50 | 0.12 | 15.82 |
| SD | 5.6 | 0.2 | 524 | 337 | 7.67 | 56.51 | 28.52 | 43.80 | 158.73 | 82.25 | 20.79 | 0.21 | 10.66 |

Note: All parameters are expressed in mg/L, except EC (µs/cm), temperature (T °C), and pH.

The minimum value of TDS was 1702 mg/L, which was greater than the irrigation water's permissible limits [30]. Freeze and Cherry [53] state that the groundwater of the CT aquifer is classified as brackish water. The water samples' pH values ranged from 7 to 7. 89 (neutral to slightly alkaline), and it did not exceed the standard limits for irrigation water [30]. The calcium concentration in all of the samples collected met irrigation water standards [30] and fell within a range of 168.33 and 340.68 mg/L. The water analysis in the Debila region had the highest $Ca^{2+}$ concentration value. Approximately 4.5% of the groundwater samples had a low concentration value of $Mg^{2+}$ and were within irrigation water's permissible limits, while the rest were above the standard limits, with an average concentration value of 109.7 mg/L [30]. The $K^+$ concentration of in all sites were above the permissible irrigation water standard, with a maximum value of 24 mg/L and a minimum value of 12 mg/L [30]. The groundwater of the CT aquifer had an acceptable concentration of $Na^+$ for irrigated agriculture, with values varying from 210 to 540 mg/L. The collected samples from the western part were more $Na^+$ enriched. Cl and $SO_4$ ions were the main dominant anions of the collected water samples, with average values of 822.25 and 697.53 mg/L, respectively. The concentrations of sulfate, chloride, and bicarbonate ions in all groundwater samples were acceptable for irrigation water [30]. In general, chemical pollution is linked with an excess of $NO_3$ ions in groundwater resources. In addition to the nitrogen cycle, other nitrate sources in groundwater come from agricultural and industrial drainage, livestock facilities, and chemical fertilizers [54]. The nitrates results revealed that 71.1% of the collected samples were greater than the irrigation water standard limit [30]. The primary concentration value of $NO_3$ ions that originate in groundwater naturally should not be over 10 mg/L. The mean concentration value of the $NO_3$ ions in the Souf Valley was 15.2 mg/L, which refers to the significant effect of anthropogenic activities in the chemical pollution of groundwater in the CT aquifer [55].

### 3.2. Groundwater Facies and Controlling Geochemical Processes

A Piper plot was developed to classify groundwater hydrochemical facies [50]. The cationic triangle showed that 15.5% of all of the collected samples (south of El-Oued and west of Debila) belonged to the $Na^+ + K^+$ class, while the remaining were in the nondominant class. In the anionic triangle, $Cl^-$ was the dominant class in 93.3% of the samples, while three water samples in the Robbah, Mih-Ouensan, and west Debila areas

were in a nondominant class. In the Piper diagram's diamond shape, the groundwater samples of the CT aquifer are divided into three main hydrochemical facies (Figure 5a). Approximately 31 samples fell within the Ca-Mg-SO$_4$ water type with permanent hardness owing to reverse ion exchange. The high salinity of the water samples was due to the elevated concentration of chloride and calcium ions, especially in the region of Debila. Six samples from El-Oued and Debila had Na$^+$-Cl$^-$ facies because evaporation was the primary factor controlling the groundwater chemistry. The remaining were in the mixed Ca$^{2+}$-Mg$^{2+}$-Cl$^-$ class zone and distributed across Hassi Khalifa, El-Oued, and Debila. A Chadha diagram was applied to verify the main dominant geochemical process controlling the groundwater chemistry in the current study [56]. Approximately 88% of the samples were in the reverse ion exchange zone. Due to the evaporation process, 12% of the samples fell in the field of seawater and was distributed in Debila, Hassi-Khalifa, and Trifaoui (Figure 5b).

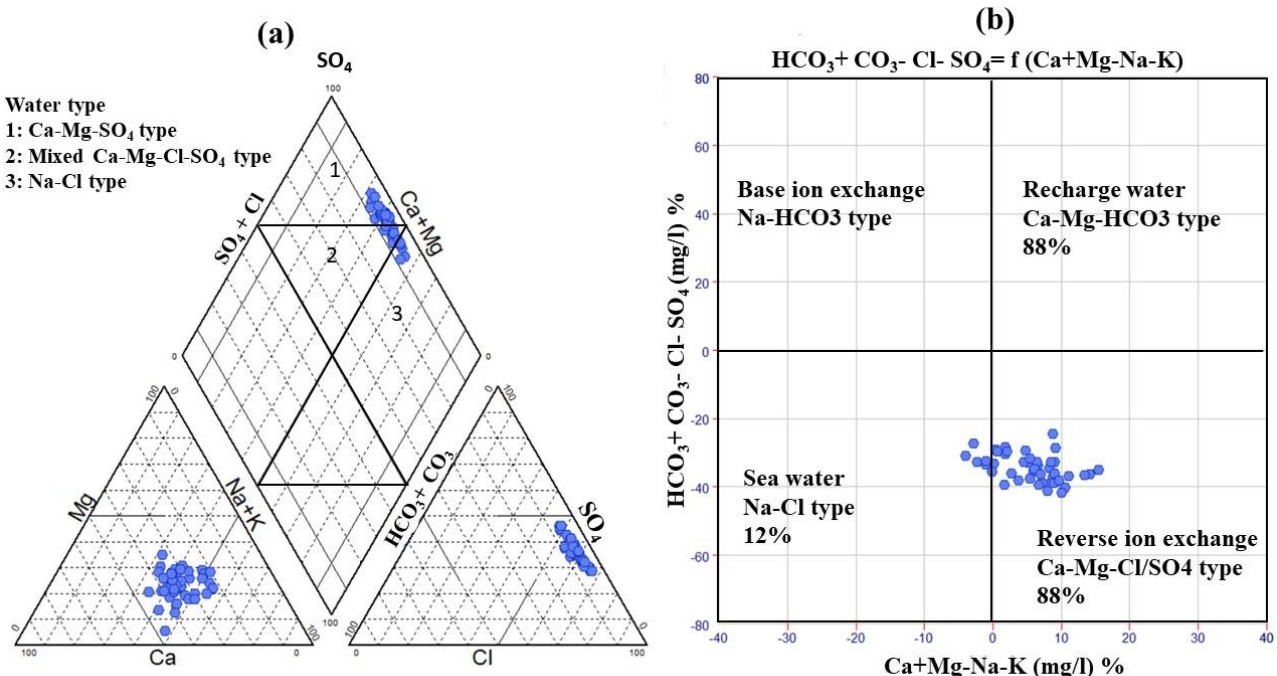

**Figure 5.** Graphical representation of the hydrochemical facies and the mechanism controlling the water chemistry: (**a**) Piper diagram; (**b**) Chadha diagram.

By applying the Gibbs plot to understand the effects of the various mechanisms that control the water chemistry, the diagram was classified into three fundamental zones (Figure 6a). The first zone is predominately precipitation that has low TDS and a high ratio of Na$^+$/(Na$^+$ + Ca$^{2+}$) and Cl$^-$/(Cl$^-$ + HCO$_3^-$). The second domain is distinguished by medium TDS and the abovementioned cation/anion ratio, which represents rock weathering. The evaporation/crystallization domain in the upper half of the Gibbs diagram with extremely high TDS is the last mechanism [51].

The diagram showed that the groundwater of the CT aquifer was controlled by the evaporation/crystallization process. The water started to precipitate the oversaturated minerals with increasing TDS, and reverse ion exchange can play a significant factor controlling the groundwater chemistry. A Durov diagram, which connects pH, TDS, and major ions, has been utilized as a visualization technique in hydrogeology by various researchers [57]. The Durov diagram can explain three major processes: mixing/dissolution, ion exchange, and reverse ion exchange (Figure 6b). With a TDS greater than 1500 mg/L, all of the samples fell within the reverse ion exchange zone, proving the previous statistical explanation.

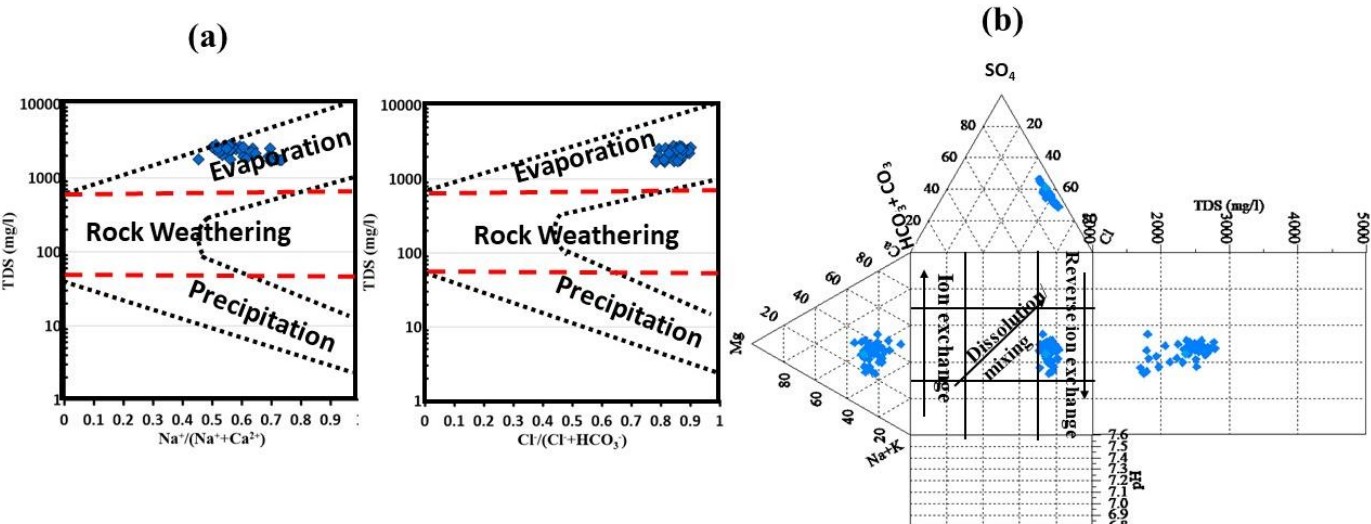

**Figure 6.** Graphical representation of the main geochemical mechanism that controls the chemistry of the water through integration of the physicochemical parameters: (**a**) Gibbs plot; (**b**) Durov diagram.

The statistical analysis was utilized to illustrate the main processes governing the chemistry of the groundwater in Souf Valley using the ratios and relationships between the major ions (Figure 7). The effect of the evaporation process in the CT aquifer can be explained using the graphical relationship between the $Na^+/Cl^-$ ratio and the EC [58]. The $Na^+/Cl^-$ ratio declined with the increase in the value of the EC as a result of the Na ion depletion caused by reverse ion exchange (Figure 7a). A Chadha diagram and various ionic plots were used to verify the reverse ion exchange effect on the water chemistry of the CT aquifer. The linearity of the relationship of $Cl^-$ and $Na^+$ demonstrated the imbalance between the two ions, as only a few groundwater samples fell on a 1:1 line graph because of a typical source (halite dissolution) (Figure 7b). The majority of the collected samples were scattered beneath the 1:1 line graph due to the fact of chloride enrichment, which is evidence of different sources of chloride ion or the removal of sodium ion from the groundwater.

Anthropogenic activities, including the drainage of excess irrigation water from cropland and waste disposal [59,60] or the atmospheric deposition of $Cl^-$ [61], can cause elevated chloride concentrations. If the $Na^+/Cl^-$ ratio is higher than one, weathering of silicate minerals could be a significant factor [62], but from the current results of the water samples, the ratio was less than one for all of the samples due to the lack of silicate weathering. The $Ca^{2+} + Mg^{2+}$ versus $Na^+ + K^+$ relationship revealed that the samples of the water were categorized into three groups, and the majority of the collected samples overlie the 1:1 line graph (Figure 7c), three samples crossed the line (1:1 line), and five samples were scattered under the line. The abundance of $Mg^{2+}$ and $Ca^{2+}$ ions over $K^+$ and $Na^+$ ions in the majority of the water samples indicates that sodium ions were replaced by $Ca^{2+}$ and $Mg^{2+}$ via the direct ion exchange process [63].

The linear relationship between the $HCO_3^- + SO_4^{2-}$ and $Ca^{2+} + Mg^{2+}$ ions (Figure 7d) revealed that most of the collected samples overlay the 1:1 line, reflecting a reverse ion exchange process. Due to the gypsum/calcite/dolomite dissolution, one sample fell on the 1:1 line graph. Reverse ion exchange is a main reason for the greater abundance of $Ca^{2+} + Mg^{2+}$ over $HCO_3^- + SO_4^{2-}$ [64]. The $Ca^{2+} + Mg^{2+}/HCO_3^-$ ratio could clarify the $Ca^{2+}$ and $Mg^{2+}$ source in the groundwater of the CT aquifer (Figure 7e). If the value of $Ca^{2+} + Mg^{2+}/HCO_3^-$ ratio was near 0.5, the magnesium and calcium ions would be derived mainly from the silicate and carbonate mineral weathering [65]. If the ratio was <0.5, the bicarbonate enrichment and/or ion exchange could be the significant factor for the magnesium and calcium ions' depletion. All of the water samples had a ratio >0.5. Because of the slightly alkaline condition of the groundwater, the depletion of $HCO_3^-$ as a cause

of this high ratio value was neglected, leaving only the reverse ion exchange as the main process accounting for all of the samples overlaying the 1:1 line (ratio = 0.5) [66].

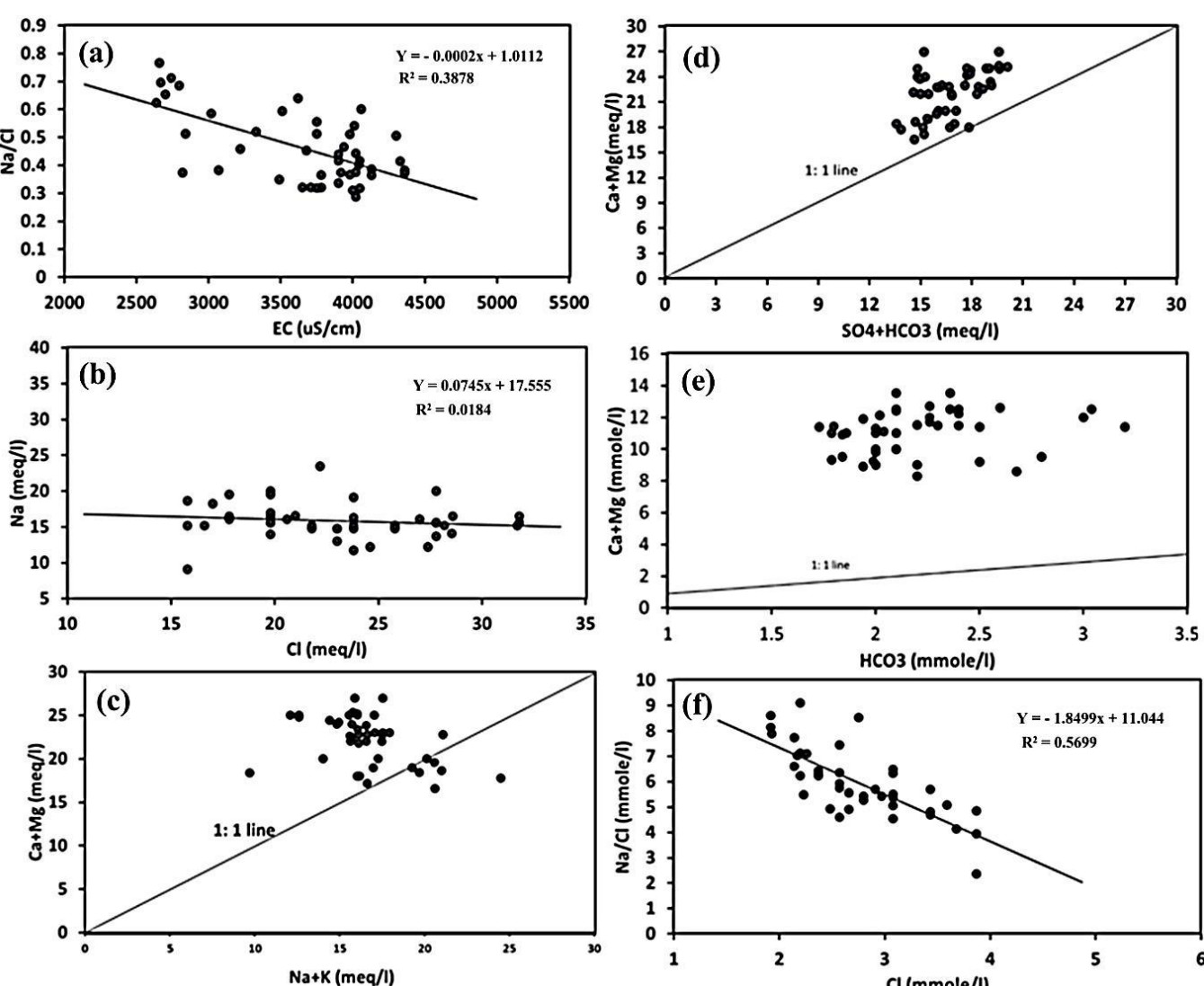

**Figure 7.** Ionic relationships between the different physicochemical parameters: (**a**) $Na^+/Cl^-$ and EC; (**b**) $Na^+$ and $Cl^-$; (**c**) $Ca^{2+} + Mg^{2+}$ and $Na^+ + K^+$; (**d**) $Ca^{2+} + Mg^{2+}$ and $SO_4^{2-} + HCO_3^-$; (**e**) $Ca^{2+} + Mg^{2+}$ and $HCO_3^-$; (**f**) $Na^+/Cl^-$ and $Cl^-$.

The value of the $Ca^{2+} + Mg^{2+}/HCO_3^-$ ratio could be useful to determine the groundwater recharge and its meteoric nature. If this ratio is less <1, the water is meteoric and there is recharge for the aquifer [67]. All of the collected samples of the CT aquifer had a remarkably large ratio, varying from 3.2 to 6.5, indicating the lack of meteoric nature as well as indicating the recharge of the groundwater. The linear graph between $Na^+/Cl^-$ and $Cl^-$ (Figure 7f) shows an inverse association, suggesting that calcium and magnesium replaced sodium caused by halite dissolution in the CT aquifer matrix (clay minerals) [64].

*3.3. Water Quality Indices for Agricultural Purposes*

The water quality, agricultural activities, and soil types play a role in deciding on the best irrigation techniques [68,69]. In terms of irrigation, multiple IWQIs were used to assess the suitability of groundwater for agriculture, including several indices. These strategies emphasize the potential soil salinization risk in addition to the negative effects of irrigation

on the characteristics of plants and the structure of soils. The data from the IWQIs and the appropriateness of the water for agriculture were statistically analyzed (Tables 4 and 5).

**Table 4.** Statistical analysis of the IWQIs values.

| Parameters | IWQI | Na% | SAR | SSP | KI | PS | PI | RSC |
|---|---|---|---|---|---|---|---|---|
| Min. | 44.76 | 31.71 | 3.01 | 30.53 | 0.44 | 21.33 | 34.02 | −26.69 |
| Max. | 56.63 | 55.35 | 6.79 | 54.08 | 1.18 | 36.49 | 58.17 | −12.80 |
| Mean | 50.78 | 43.07 | 4.85 | 41.78 | 0.74 | 29.60 | 45.65 | −20.44 |
| SD | 2.28 | 5.62 | 0.82 | 5.56 | 0.17 | 3.73 | 5.68 | 3.58 |

Notes: SD: standard deviation; Min.: minimum; mean: average; Max.: maximum.

**Table 5.** Classification of the different IWQIs for irrigation according to the documented references.

| Index | Range | Water Category | Number of Samples (%) |
|---|---|---|---|
| IWQI | 0–40 | Severe restriction | 0 (0%) |
| | 40–55 | High restriction | 43 (93.6%) |
| | 55–70 | Moderate restriction | 2 (4.4%) |
| | 70–85 | Low restriction | 0 (0.0%) |
| | 85–100 | No restriction | 0 (0.0%) |
| Na% | 40–60 | Permissible | 29 (64.4%) |
| | 20–40 | Good | 16 (35.6%) |
| | <20 | Excellent | 0 (0.0%) |
| SAR | >26 | Unsuitable | 0 (0.0%) |
| | 18–26 | Doubtful or fairly poor | 0 (0.0%) |
| | 10–18 | Good | 0 (0.0%) |
| | <10 | Excellent | 45 (100%) |
| SSP | >60 | Unsafe | 0 (0.0%) |
| | <60 | Safe | 45 (100%) |
| KI | >1 | Unsuitable | 41 (91.2%) |
| | <1 | Good | 4 (8.8%) |
| PS | >5 | Injurious to unsatisfactory | 45 (100%) |
| | 3–5 | Good to injurious | 0 (0.0%) |
| | <3 | Excellent to good | 0 (0.0%) |
| PI | <25% | Unsuitable—Class III | 0 (0.0%) |
| | 25–75% | Good—Class II | 45 (100%) |
| | >75% | Good—Class I | 0 (0.0%) |
| RSC | >2.5 | Unsuitable | 0 (0.0%) |
| | 1.25–2.5 | Marginal | 0 (0.0%) |
| | <1.25 | Safe | 45 (100.0%) |

### 3.4. Irrigation Water Quality Index (IWQI)

The groundwater assessment for irrigation use through IWQI requires using individual chemical indices [70,71] or several indices combined [31,72]. Although the evaluation of irrigation-related groundwater quality depending on individual parameters is useful, the combined indices give more effective information for decision makers. Five hazard groups were utilized to estimate the water safety for irrigation purposes [73]. The resulting IWQI values varied from 44.76 to 56.63, with a mean value of 50.78 (Table 4), and the IWQI available classification revealed that a large percentage of the groundwater samples (93.6%) fell in a high restriction class, while 4.4% of the collected samples were in the moderate restriction class (Table 5). The overall index map illustrates the appropriateness of the water for irrigation depending on the main physical and chemical parameters (Figure 8a). This map has the ability to estimate groundwater validation for irrigation. The deterioration of the water quality based on the IWQI values was reported in the northeast part, near Debila and El-Oued, as a result of anthropogenic activities and geogenic sources.

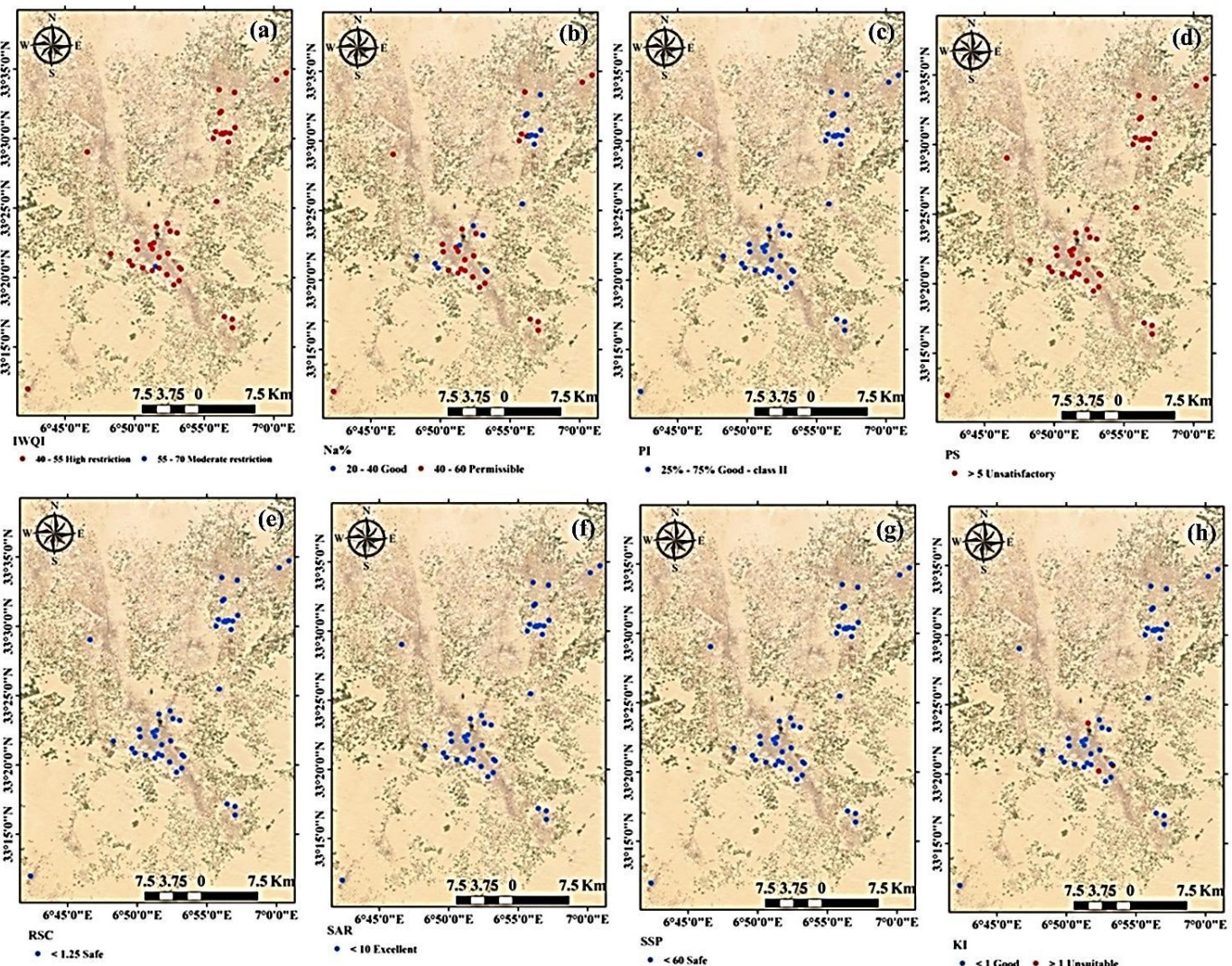

**Figure 8.** Spatial distribution maps of IWQIs in Souf Valley: (**a**) IWQI; (**b**) Na%; (**c**) SAR; (**d**) SSP; (**e**) KI; (**f**) PS; (**g**) PI; (**h**) RSC.

### 3.5. Sodium Percentage (Na%)

Due to the high sodium ion concentrations in water for irrigation reacting with the soil particles and lessening its permeability, Na% is applied to evaluate the fitness of groundwater resources for irrigation [32]. Once a high $Na^+$ concentration value exists in water, clay minerals absorb it and release $Mg^{2+}$ and $Ca^{2+}$ ions. The exchange of $Na^+$ in water for $Ca^{2+}$ and $Mg^{2+}$ in soil decreases the permeability, causing a reduction in the soil infiltration. As a consequence, water and air flow are restricted under wet conditions, and soils generally harden during dry conditions [74].

The calculated Na% for the CT aquifer varied from 31.71 to 55.35%, with an average value of 43.07% (Table 4). According to the sodium percent [23], the water in the study region had a Na% value ranging from 31.71% to 55.35%, indicating a good to permissible irrigation quality (Figure 8b).

### 3.6. Sodium Adsorption Ratio (SAR)

The SAR index is utilized in irrigation as an indicator to refer to the capability of the soil to remove $Mg^{2+}$ and $Ca^{2+}$ ions and absorb $Na^+$ ions from the groundwater at ion exchangeable sites, eventually causing soil particle dispersion and a decline in the infiltration capacity [75,76]. Despite irrigation water's high salinity, it can benefit the soil

composition by enhancing its infiltration rate; it causes more drought stress to the plants. If the salinity of the irrigation water is very high, crops and plants must undergo an energy-intensive process to obtain water from the soil. The samples of water were plotted on a USSL diagram [33] to investigate how the water quality affects the crops' traits and yield. This diagram represents the relationship of the EC and SAR and categorizes them into various classes (Figure 9). Forty-four water samples were classified as C4-S2 (very high salinity–medium SAR), and one sample was classified as C4-S1 (very high salinity–low SAR). The maximum SAR value in all samples was less than 10 (Figure 8c).

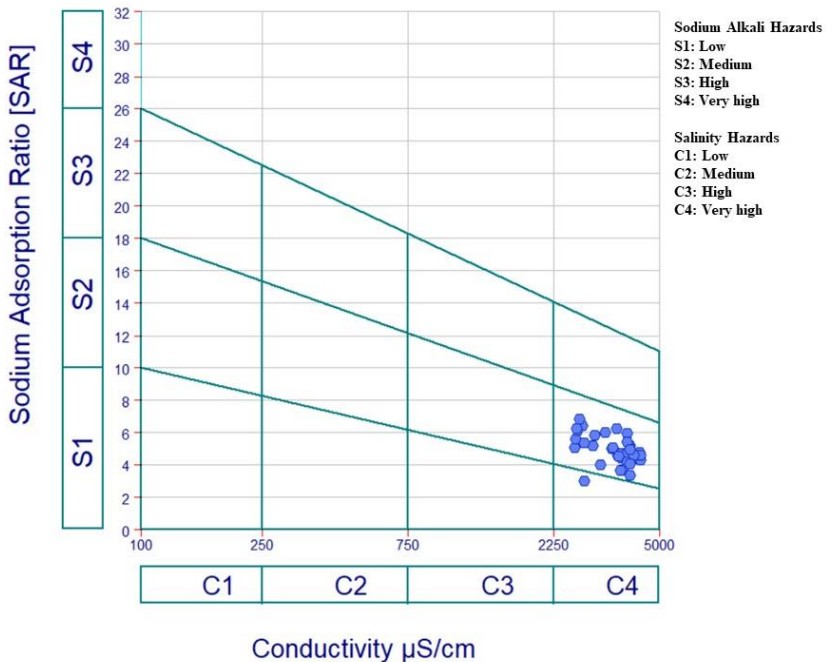

**Figure 9.** USSL diagram for irrigation purposes.

The findings showed that the high salinity of the irrigation water harmed the plants, while there was no impact on the soil infiltration capacity because of the low to medium SAR value (Figure 9), and there was no need for the use of calcium fertilizers. The optimum use-related management for the use of groundwater for irrigation is to select crops and plants that are resilient to the irrigation water's high salinity.

### 3.7. Soluble Sodium Percentage (SSP)

The SSP was applied to calculate the salinity by comparing the sodium concentrations to the calcium and magnesium concentrations. A high $Na^+$ concentration in the water compared to $Ca^{2+}$ and $Mg^{2+}$ causes toxicity materials, which contribute to damaged leaves and dead plant tissues [77]. According to the SSP results, 100% of the groundwater samples had an excellent and safe quality for irrigation, with values ranging from 30.53 to 54.08 (Figure 8d).

### 3.8. Kelly Index (KI)

The KI was calculated to determine whether the groundwater was suitable for irrigation usage [78]. From 0.44 to 1.18, with a mean of 0.74, the value of KI was recorded. Values of KI greater than one (KI > 1) indicate that the water contains an excess of sodium, whereas a value below one (KI 1) denotes that the water is appropriate for irrigation [34,79]. Based on the KI results, 91.2% of the total samples were classified as unsuitable, while the remaining samples (8.8%) were classified as good and appropriate for irrigation (Table 5 and Figure 8e).

### 3.9. Potential Salinity (PS)

The PS uses the concentration of $Cl^-$ and half of the $SO_4^{2-}$ concentration to assess if groundwater is suitable for use in irrigation. The PS values are generally classified into two categories: unsuitable (>3) and suitable (<3) for irrigated agriculture [24]. The obtained results showed that the PS values ranged from 21.33 to 36.49, with an average value of 29.60. According to the PS, the study area's groundwater samples were all unsuitable for irrigation (Figure 8f).

### 3.10. Permeability Index (PI)

The PI is an important irrigation water quality parameter, because continuous irrigation water utilization affects the soil permeability, which is governed by soil components, such as sodium, magnesium, calcium, and bicarbonate ions. According to Table 5, the PI value in all water samples was found to be in the good quality range (34.02–58.17), as shown in Figure 8g.

### 3.11. Residual Sodium Carbonate (RSC)

Another aspect influencing irrigation water quality is excessive carbonates and bicarbonates in relation to $Ca^{+2}$ and $Mg^{+2}$ ions, which can reduce irrigation water quality by precipitating alkali metals, primarily $Mg^{+2}$ and $Ca^{+2}$. The SAR value and sodium ion concentrations may both rise as a result of the precipitation of $Ca^{+2}$ and $Mg^{+2}$ as carbonate minerals. [35]. High RSC has the potential to deteriorate the soil's physical qualities by causing the dissociation of organic matter, which eventually results in a black stain on the soil's surface after drying [80,81].

The RSC was computed to determine the possible precipitation of $Ca^{2+}$ and $Mg^{2+}$ on the particles of the soil's surface. The values of the RSC index in groundwater is reported to be high in areas that are dry to semi-dry, causing soil sodification and soil salinization [82]. According to the RSC values, the groundwater was divided into three categories (Figure 8h). Irrigation water with an RSC greater than 2.5 is not appropriate for irrigation, whereas water with an RSC less than 1.25 is good, and water with an RSC that ranges from 1.25 and 2.5 is doubtful for use in irrigation [35]. In the current study, all groundwater samples had an RSC value < 1.25, demonstrating that the groundwater was suitable and safe for irrigation uses (Figure 8h).

### 3.12. Performance of the Support Vector Machine Regression Based on Physicochemical Parameters for Predicting the Irrigation Water Quality Indices

Several researchers have investigated methods for reducing the subjectivity of established water quality index technology, which has been proven to be a more accurate and precise essential tool for reliable weighing systems by assigning weights to critical ions based on entropy [83]. Water quality research, on the other hand, necessarily requires a significant amount of data collection, laboratory analysis, data management, and testing [84]. As a consequence of the computation's subjectivity, the WQIs' interpretation of the results contained inconsistencies. It is possible to identify a subset of features that have high predictive and discriminative potential using methods for feature selection based on models [85]. By minimizing overfitting and eliminating pointless features, this strategy can enhance the model performance. The original feature representation should be kept, because it provides a number of benefits on top of improving the interpretability [86]. In the disciplines of modeling and prediction, there is an increasing need for feature selection algorithms [87].

Mathematical techniques can be used to estimate the IWQIs of water sites with accuracy. These techniques, however, are difficult to apply to evaluate IWQIs, because they require a number of mathematical equations to convert a sizable amount of data on water characterization into a single value that characterizes the water quality levels and reflects the overall water quality level. This study evaluated the SVMR model to predict the IWQIs based on the numerous response factors of the chemical parameters. The SVMR model

was used in this work to anticipate the IWQIs based on various parameters, as shown in Table 6, because it is fast and does not require several more steps to construct the IWQIs.

**Table 6.** Outcomes of the calibration and validation models of the SVMR of the association between the observed and predicted IWQI, Na%, SAR, SSP, KI, PS, PI, and RSC of the groundwater quality.

| Variable | Calibration | | Validation | |
| --- | --- | --- | --- | --- |
| | $R^2$ | RMSE | $R^2$ | RMSE |
| IWQI | 0.93 *** | 0.85 | 0.90 *** | 0.92 |
| Na% | 0.97 *** | 1.25 | 0.92 *** | 1.37 |
| SAR | 0.94 *** | 0.22 | 0.92 *** | 0.27 |
| SSP | 0.96 *** | 1.21 | 0.95 *** | 1.42 |
| KI | 0.97 *** | 0.04 | 0.96 *** | 0.05 |
| PS | 0.95 *** | 1.13 | 0.94 *** | 1.20 |
| PI | 0.90 *** | 1.79 | 0.88 *** | 2.09 |
| RSC | 0.94 *** | 1.02 | 0.92 *** | 1.15 |

Note: *** Statistically significant at $p \leq 0.001$.

The SVMR model was utilized to more accurately assess eight IWQIs relying on the $R^2$ and RMSE values (Table 6) and the slope (Figures 10 and 11). With $R^2$ values ranging from 0.90 to 0.97, the SVMR model achieved robust estimates for eight IWQIs in the Cal. datasets. Moreover, the SVMR model produced accurate estimations for eight IWQIs in the Val. datasets, with the $R^2$ ranging from 0.88 to 0.95. The validation model's RMSE values for eight IWQIs, including IWQI, Na%, SAR, SSP, KI, PS, PI, and RSC, as shown in Table 6, were 0.92, 1.37, 0.27, 1.42, 0.05, 1.20, 2.09, and 2.09, respectively. Figures 10 and 11 show the SVMR based association of the eight IWQIs. Furthermore, these figures showed a reasonable slope of the linear relationship between the predicted and measured validation model values for every index, with IWQI having the highest slope (1.0128) and PS having the lowest slope (0.8318). There was no overfitting or underfitting in the datasets used to measure, calibrate, and validate the SVMR models of the eight IWQIs. As an outcome, the models applied in this study had sufficient accuracy and performed well when forecasting the IWQIs. These findings are consistent with those in [88], which found that the principal component regression produced precise and powerful models that predicted the IWQIs, with $R^2$ values ranging from 0.48 to 0.99. Based on four parameters, including temperature, turbidity, pH, and TDS, Ahmed et al. [89] discovered that supervised machine learning with multiple linear regression could be used to estimate the water quality index of surface waters with an $R^2$ value of 0.66. Multiple linear regression models with $R^2$ values that reached 0.64 were discovered by Chen and Liu [90] to be useful for estimating water quality indicators, such as dissolved oxygen, total phosphorus, and chlorophyll disc depth.

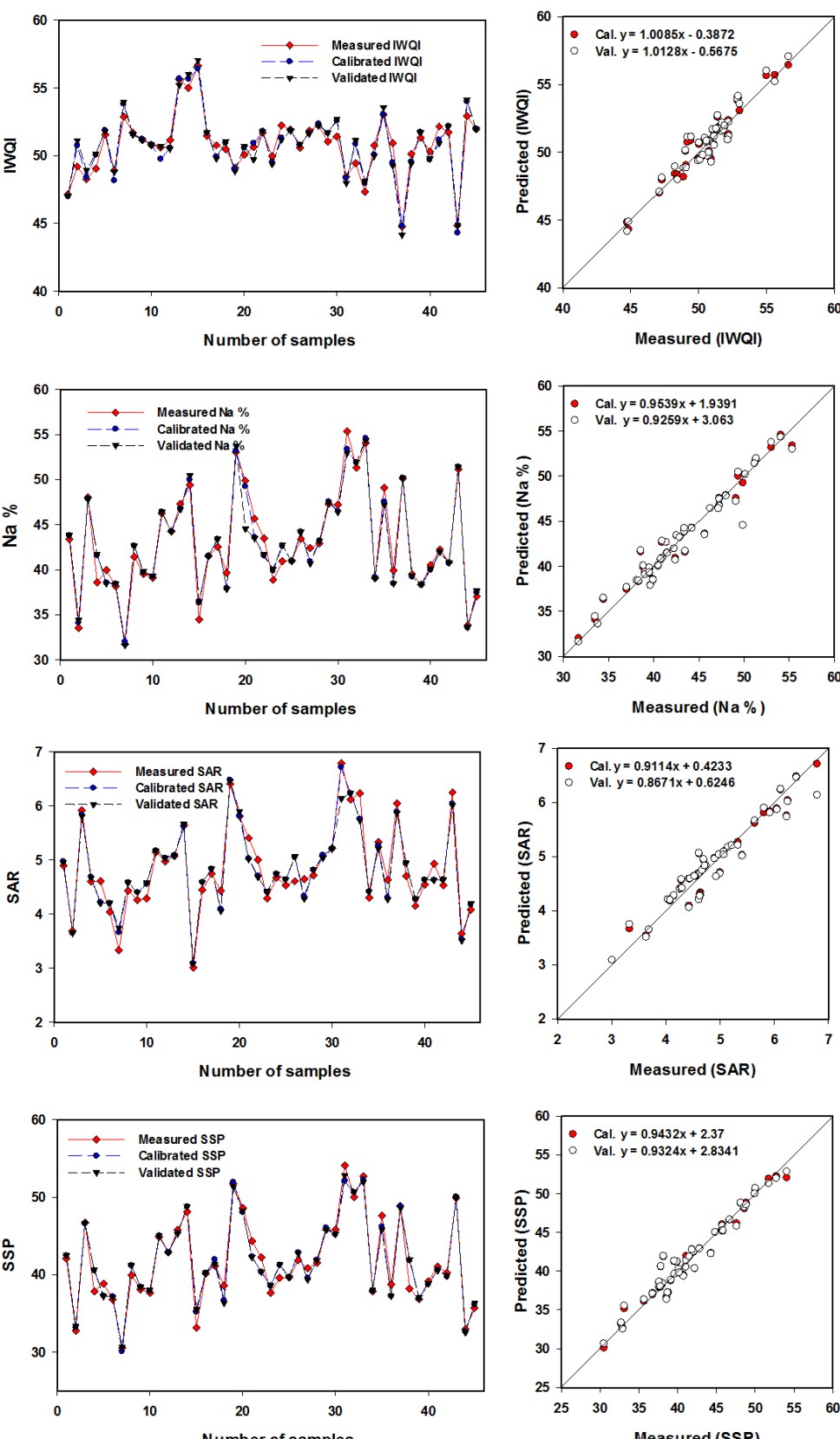

**Figure 10.** Relationships between the measuring, calibration, and validating datasets of the output (IWQI, Na%, SAR, and SSP) using the SVMR models. The results of the statistical analysis are displayed in Table 6.

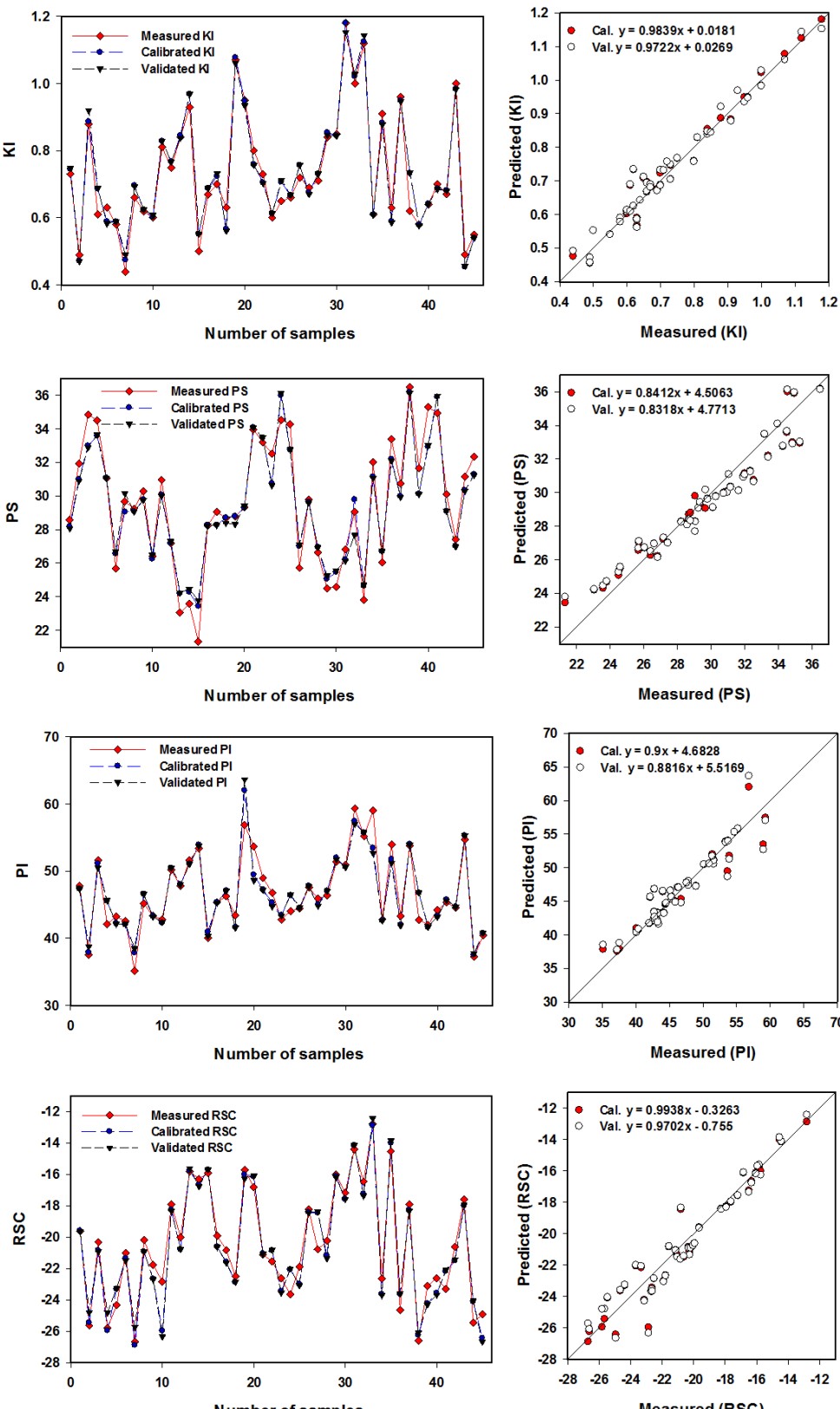

**Figure 11.** Relationships between the measuring, calibration, and validating datasets of the output (KI, PS, PI, and RSC) using the SVMR models. The results of the statistical analysis are displayed in Table 6.

## 4. Conclusions

This research study in the Souf Valley of the northeast Algerian Sahara investigated the appropriateness of the CT aquifer for agricultural purposes. Physicochemical properties, IWQIs, SVMR, and geographic information systems (GIS) techniques were carried out to identify groundwater hydrogeochemical facies and their controlling mechanisms. The regional flow direction of the groundwater is from the southwest to the northeast. There was a rapid decrease in the piezometric level, especially in the El-Oued, Trifaoue, and Baydah regions due to the fact of over pumping for irrigation purposes.

This study provides insights into the appropriateness of the CT aquifer in Algeria for irrigation purposes. To detect the groundwater hydrogeochemical types and their controlling mechanisms, the physicochemical properties, IWQIs, SVMR, and GIS techniques were used. The physicochemical parameters obtained revealed that the hydrochemical facies were Na-Cl, and Ca-Mg-Cl/SO$_4$, which indicates that the significant hydrochemical processes that govern the chemistry of the groundwater in the complex terminal aquifer were evaporation/crystallization, reverse ion exchange, and rock–water interaction.

The IWQI, SAR, KI, and PS of the groundwater in the Souf Valley revealed that the water was categorized for irrigation purposes into high restriction (93.6%), permissible (64.4%), unsuitable (91.2%), and injurious to unsatisfactory (100%), respectively, while Na%, SSP, PI, PI, and RSC revealed that all of the groundwater was excellent, safe, and good-class II for irrigation. Combining the physicochemical parameters, IWQIs, GIS, and ML approaches is, therefore, efficient and provides a comprehensive image of groundwater fitness for irrigation purposes and its controlling factors. Furthermore, the technique proposed in this paper could be studied further to raise its reliability for groundwater under different conditions, and it enables decision makers to integrate different technologies for water quality management and planning. As a result, in this study, we attempted to overcome the limitations of the traditional methods by forecasting the groundwater quality for irrigation purposes using ML models under extensive salinization.

**Author Contributions:** M.H.E., analyzing and visualizing the data, interpreting and writing the first draft of the article; M.G., S.E., M.E. and A.E.-D.O., calculation of indices and writing the draft; A.A.T., quantitative and qualitative data collection, writing the introduction, and the previous work; H.H. and F.S.M., mapping using GIS and the distribution maps of the different parameters and indices; S.E., machine learning and the modeling and prediction of the water quality indices; S.P. and A.K., editing and correcting the scientific and language errors. All authors have read and agreed to the published version of the manuscript.

**Funding:** This research was funded through the Large Groups Project under grant number L.G.P. 2/138/43.

**Data Availability Statement:** All data are provided in the tables and figures.

**Acknowledgments:** The authors extend their appreciation to the King Khalid University for funding this work through the Large Groups Project under grant number L.G.P. 2/138/43.

**Conflicts of Interest:** The authors declare no conflict of interest.

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
