# Peer review of "Evaluation of Groundwater Quality for Irrigation in Deep Aquifers Using Multiple Graphical and Indexing Approaches Supported with Machine Learning Models and GIS Techniques, Souf Valley, Algeria"

_water, doi:10.3390/w15010182_

Round 1

Reviewer 1 Report

Line 33: Kelley Kelly

Line 34: “Permeability index” has been duplicated. Please remove.

Lines 38-41: The results are contradictory. Please justify.

Abstract can end with some implications of the findings in a broader context. For example, what can others learn from your investigation? How can they apply your findings to their own plain/aquifer?

Keywords: Please do not use abbreviation here.

Lines 51-55: Please support this section with appropriate references. It would be better to use the studies conducted in (semi)arid regions such as “Iran’s groundwater hydrochemistry”.

Lines 56-70: The problem mentioned is as same as the problem that exists in (semi)arid regions of the world such as Iran. I would like to see some refereed document hear to increase the readability of your manuscript. I would suggest: “Anthropogenic depletion of Iran’s aquifers”.

Lines 78-81: Please add the original references for USSL, Wilcox, and Doneen.

Lines 87-92: Please add the original references for the IWQIs used.

Can you please add a legend to Figure 1?

How did you create Figure 3? Need a reference.

Section 2.3.3: Please elaborate this section. You need to give more information about the model used (optimization technique used, kernel type, …). Please see “An Efficient Data Driven-Based Model for Prediction of the Total Sediment Load in Rivers”.

Line 217: discissions discussion

Author Response

Reviewer 1

Manuscript title: Evaluation of groundwater quality for irrigation in deep aquifers using multiple graphical and indexing approaches supported with machine learning models and GIS techniques, Souf valley, Algeria

We greatly appreciate your critical observations as well as your constructive and helpful comments. We hope that we could address your questions/comments by the explanations and revisions made in the manuscript. We believe that the manuscript is substantially improved after making the suggested revisions.

  1. Line 33: Kelley → Kelly

Response: We greatly appreciate your critical observations. The word "Kelley" was corrected to "Kelly" in all manuscript as your suggestion (Line 33 – Page 1).

  1. Line 34: “Permeability index” has been duplicated. Please remove.

Response: We greatly appreciate your critical observations. The duplicated of "Permeability index" was removed from the abstract section as your suggestion (Line 33 – Page 1).

  1. Lines 38-41: The results are contradictory. Please justify.

Response: Many thanks for this comment. The results in the abstract section were justified as your suggestion (Lines 38 to 41 – Page 2).

  1. Abstract can end with some implications of the findings in a broader context. For example, what can others learn from your investigation? How can they apply your findings to their own plain/aquifer?

Response: Many thanks for this comment. The implications of the findings were added at the end of abstract section as your suggestion (Lines 50 to 53– Page 2).

  1. Keywords: Please do not use abbreviation here.

Response: We greatly appreciate your critical observations. The abbreviation was deleted from the keywords as your suggestion (Line 55 – Page 2).

  1. Lines 51-55: Please support this section with appropriate references. It would be better to use the studies conducted in (semi) arid regions such as “Iran’s groundwater hydrochemistry”.

Response: Many thanks for this comment. This section was supported with appropriate references “Iran’s groundwater hydrochemistry” as your suggestion (Lines 66 to 71 – Page 2).

  1. Lines 56-70: The problem mentioned is as same as the problem that exists in (semi) arid regions of the world such as Iran. I would like to see some refereed document hear to increase the readability of your manuscript. I would suggest: “Anthropogenic depletion of Iran’s aquifers”.

Response: Many thanks for this comment. This section was supported with appropriate references “Anthropogenic depletion of Iran’s aquifers” as your suggestion (Lines 66 to 74 – Pages 2, 3).

  1. Lines 78-81: Please add the original references for USSL, Wilcox, and Doneen.

Response: Many thanks for this comment. The original references for USSL, Wilcox, and Doneen were added as your suggestion (Line 95 – Page 3).

  1. Lines 87-92: Please add the original references for the IWQIs used.

Response: Many thanks for this comment. The original references for the IWQIs were added as your suggestion (Line 104 – Page 3).

  1. Can you please add a legend to Figure 1?

Response: We greatly appreciate your critical observations. The legend was added to Figure 1 as your suggestion (Line 145– Page 5).

  1. How did you create Figure 3? Need a reference.

Response: Many thanks for this comment. Figure 3 was reconstructed according to the obtained data by interpolating points to raster using the Inverse Distance Weighting interpolation method (IDW) using Surfer software package v. 15.0 (Line 169 – Page 7).

  1. Section 2.3.3: Please elaborate this section. You need to give more information about the model used (optimization technique used, kernel type, …). Please see “An Efficient Data Driven-Based Model for Prediction of the Total Sediment Load in Rivers”.

Response: Many thanks for this comment. More information about the model used (optimization technique used, kernel type, …) were added as your suggestion (Lines 224 to 243 – Pages 9, 10).

  1. Line 217: discissions → discussion

Response: We greatly appreciate your critical observations. The word "Discussions" was corrected to "discussion" as your suggestion (Line 261 – Page 12).

Reviewer 2 Report

Thank you for the opportunity to review your manuscript.  The manuscript provides a good overview of aspects of irrigation water quality indices.  However more information is needed regarding the machine learning method used and why it was used as opposed to others.  Most of the figures seem reasonable, although perhaps some could be summarized or only include representative examples of the IWGI relationships.  Figure 3 appears to have been generated using a contouring program; however, numerous automated contouring artifacts are present, contours exist outside of available data, and "bulls-eye" patterns dominate.  I suggest hand-contouring the data using any soft data available information regarding confinement conditions.  The discussion of controls on water composition seem to focus on generalities derived from the indices rather than process-oriented interpretations.  Please consider discussing more of the mineralogical (dissolution and precipitation) processes as well as mixing and evaporative concentration.  The use of the machine language regression of the IWQIs does not address the lack of independence of the indices and concerns expressed by other researchers.  More discussion needs to be focused in this area to make this manuscript of the quality that it can be.

Author Response

Reviewer 2

Manuscript title: Evaluation of groundwater quality for irrigation in deep aquifers using multiple graphical and indexing approaches supported with machine learning models and GIS techniques, Souf valley, Algeria

The manuscript provides a good overview of aspects of irrigation water quality indices.  However more information is needed regarding the machine learning method used and why it was used as opposed to others.  Most of the figures seem reasonable, although perhaps some could be summarized or only include representative examples of the IWGI relationships. Figure 3 appears to have been generated using a contouring program; however, numerous automated contouring artifacts are present, contours exist outside of available data, and "bulls-eye" patterns dominate. I suggest hand-contouring the data using any soft data available information regarding confinement conditions. The discussion of controls on water composition seems to focus on generalities derived from the indices rather than process-oriented interpretations. Please consider discussing more of the mineralogical (dissolution and precipitation) processes as well as mixing and evaporative concentration. The use of the machine language regression of the IWQIs does not address the lack of independence of the indices and concerns expressed by other researchers.  More discussion needs to be focused in this area to make this manuscript of the quality that it can be. 

We greatly appreciate your critical observations as well as your constructive and helpful comments. We hope that we could address your questions/comments by the explanations and revisions made in the manuscript. We believe that the manuscript is substantially improved after making the suggested revisions.

  1. Lines 39, 40 – Page 2. The percentages are not useful because it is unclear to what they pertain.

Response: Many thanks for this comment. The percentages of IWQIs results were removed from the abstract section as your suggestion (Lines 38 to 41– Page 2).

  1. Line 57, Page 2. risks or actual pollution?

Response: Many thanks for this comment. This sentence was modified as your suggestion (Lines 66 to 68 – Page 2).

  1. Line 62 – Page 2. I am not familiar with these types of aquifers

Response: Many thanks for this comment. This sentence was re-written to more clarify (Lines 72 to 74 – Pages 2, 3).

  1. Line 66 – Page 2. The most concentrated areas of production wells with over-abstraction is in the El-Oued and Debila regions. What is used to determine over abstraction?

Response: Many thanks for this comment. This sentence was removed from the introduction section (Lines 80 to 84 – Page 3).

  1. Line 72 – Page 2. "Therefore, irrigation water's hydrochemical elements can have a detrimental influence on crop productivity and soil degradation".

Response: We greatly appreciate your critical observations. This sentence was modified to be "Therefore, some irrigation water's hydrochemical elements can have a detrimental influence on crop productivity and soil degradation" as your suggestion (Line 86 – Page 3).

  1. Lines 79, 80 – Page 3. Ref. [25] does not seem to have bearing on the question of interest.

Response: We greatly appreciate your critical observations. The appropriate reference was added according to the relevant methods as your suggestion (Line 104 – Page 3).

  1. Line 91 – Page 3. The IWQIs, including the irrigation water quality index (IWQI), sodium adsorption ratio (SAR), sodium percentage (Na%), soluble sodium percentage (SSP), Kelly index (KI), permeability index (PI), potential salinity (PS), and residual sodium carbonate index (RSC), have been widely used to categorize the irrigation appropriateness of groundwater resources, which assist in determining the infiltration rate of the formations. "not sure this makes much sense but perhaps as the solutes concentrate in soil they change permeability".

Response: We greatly appreciate your critical observations. This sentence was modified as your suggestion (Lines 102 to 103 – Page 3).

  1. Line 94 – Page 3. Change the words "to separate the" to "separation of"

Response: We greatly appreciate your critical observations. The words "to separate the" was changed to "separation of" as your suggestion (Line 107 - Page 3).

  1. Line 99 – Page 3. Cited ref. does not appear to be ML study.

Response: We greatly appreciate your critical observations. The cited reference was corrected according to the relevant ML study as your suggestion (Line 113 - Page 4).

  1. Lines 111, 112 – Page 4. Does not appear to be complete sentence.

Response: We greatly appreciate your critical observations. The sentence was completed as your suggestion (Lines 124 to 126 – Page 4).

  1. Line 131 – Page 4. Change "Complex terminal" to "CT".

Response: We greatly appreciate your critical observations. The abbreviation "CT" was used as your suggestion (Line 149 – Page 5).

  1. Line 135 – Page 5. The thickness CT aquifer

Response: We greatly appreciate your critical observations. This sentence was corrected from "The thickness CT aquifer" to "The thickness of the CT aquifer" as your suggestion (Line 153 – Page 5).

  1. Line 139 – Page 5. chaotic potentiometric surface map - needs to be re-drawn by hand

Response: We greatly appreciate your critical observations. Figure 3 was re-drawn according to the data available information (Line 168 – Page 7).

  1. Line 142 – Page 5. Where are the other aquifers?

Response: Many thanks for this comment. The other aquifers were presented in Figure 2 as your suggestion (Line 164 – Page 6).

  1. Line 144 – Page 5. Please hand contour and show control points. This is not a publishable potentiometric surface map.

Response: Many thanks for this comment. The potentiometric surface map (Fig. 3) was re-drawn according to the data available information as your suggestion (Line 168 – Page 7).

  1. Line 148 – Page 6. Change word "points" to "samples"

Response: Many thanks for this comment. The word "Points" was changed to "samples" as your suggestion (Line 174 – Page 7).

  1. Line 162 – Page 6. CBE = (Denominator should be divided by 2).

Response: Many thanks for this comment. The formula was modified as your suggestion (Line 190 – Page 8).

  1. Line 189 – Page 7. The largely limited of parameter i by factor j. (rephrase – unclear).

Response: Many thanks for this comment. This sentence was rephrasing as your suggestion (Line 216 – Page 9).

  1. Line 191 – Page 7. Ranging from 1 to k.ij. (Is something missing?)

Response: We greatly appreciate your critical observations. This sentence was modified as your suggestion (Line 218 – Page 9).

  1. Line 193 – Page 7. Need more theoretical information about SVMR models - what are they, how are they calculated, why SVMR instead of other machine language models?

Response: Many thanks for this comment. More theoretical information about SVMR models and calculations were added in materials and methods section as your suggestion (Lines 224 to 243 – Pages 9, 10).

  1. Line 219 – Page 9. What are the deep aquifers?

Response: Many thanks for this comment. The word of deep aquifer was changed to complex terminal (CT) aquifer as your suggestion (Line 263 – Page 12).

  1. Line 226 – Page 9.

Response: We greatly appreciate your critical observations. The word "altering" was changed to "alter" as your suggestion (Line 270 – Page 12).

  1. Line 227 – Page 9.

Response: We greatly appreciate your critical observations. The word "were" was changed to "are" as your suggestion (Line 270 – Page 12).

  1. Lines 287, 288 – Page 11.

Response: Many thanks for this comment. This sentence was modified as your suggestion (Lines 320 to 326 – Page 14).

  1. Line 302 – Page 12. Waters outside the fields - excess HCO3- ?

Response: We greatly appreciate your critical observations. Figure 6a (Gibbs plot) was modified according to the obtained data as your suggestion (Line 339 – Page 15).

  1. Line 305 – Page 12.

Response: We greatly appreciate your critical observations. The word "effect" was changed to "effect of" as your suggestion (Line 242 – Page 15).

  1. Line 308 – Page 12. Unclear how these diagrams can illustrate this?

Response: Many thanks for this comment. This sentence was removed from the manuscript.

  1. Line 360 – Page 15. (Table 6) unclear which indices contribute to the IWQI category

Response: Many thanks for this comment. The title of Table 6 was modified to be more clarify as your suggestion (Line 397 – Page 18).

  1. Line 367 – Page 15.

Response: We greatly appreciate your critical observations. The word "hazards" was changed to "hazard" as your suggestion (Line 403 – Page 18).

  1. Line 411 – Page 18.

Response: Many thanks for this comment. The word "condition" was changed to "fertilizers" as your suggestion (Line 445 – Page 20).

  1. Line 461 – Page 19.

Response: We greatly appreciate your critical observations. The title was modified as your suggestion (Line 493 – Page 22).

  1. Lines 463 to 469 – Page 19. One concern is that the indices are not independent of one another. Each providing valid evaluation, but in sum interrelated

Response: Many thanks for this comment. The corrlation matrix between the all indices was established. As we see some of has poor correlation, other moderate correlation and another has strong relationship. But any way the SVMR models were performed well to predicted all IWQIs.  more information was added under section  (3.12. Performance of support vector machine regression  based on physicochemical parameters for predicting irrigation water quality indices ).

IWQI

SAR

Na%

SSP

KI

PS

PI

RSC

IWQI

1

SAR

-0.52

1

Na%

-0.39

0.97

1

SSP

-0.40

0.97

0.99

1

KI

-0.41

0.97

0.99

0.99

1

PS

-0.21

-0.12

-0.32

-0.31

-0.33

1

PI

-0.37

0.95

0.99

0.99

0.99

-0.38

1

RSC

-0.08

0.65

0.82

0.81

0.82

-0.71

0.86

1

  1. Line 471 – Page 19. Both models were used in this work to anticipate IWQIs based on various parameters (which model? Unclear)

Response: We greatly appreciate your critical observations. This sentence was modified as your suggestion (Lines 516 to 517 – Page 22).

  1. Line 475 – Page 19.

Response: We greatly appreciate your critical observations. The word "R2" was modified to "R2" as your suggestion (Line 521 – Page 22).

  1. Line 502 – Page 23. Which is this in your discussion, CT?

Response: Many thanks for this comment. This sentence was modified as your suggestion (Line 555 – Page 26).

  1. Line 508 – Page 23.

Response: Many thanks for this comment. The word "revealed" was changed to "indicate" as your suggestion (Line 565 – Page 26).

Round 2

Reviewer 1 Report

The authors have properly responded to my comments/suggestions. My suggestion is acceptance. Congratulations to the Authors.